# The Interplay between Angiopoietin-Like Proteins and Adipose Tissue: Another Piece of the Relationship between Adiposopathy and Cardiometabolic Diseases?

**DOI:** 10.3390/ijms22020742

**Published:** 2021-01-13

**Authors:** Simone Bini, Laura D’Erasmo, Alessia Di Costanzo, Ilenia Minicocci, Valeria Pecce, Marcello Arca

**Affiliations:** Department of Translational and Precision Medicine, Sapienza University of Rome, Viale del Policlinico 155, 00185 Rome, Italy; alessia.dicostanzo@uniroma1.it (A.D.C.); ilenia.minicocci@uniroma1.it (I.M.); valeria.pecce@uniroma1.it (V.P.); marcello.arca@uniroma1.it (M.A.)

**Keywords:** adipose tissue, adiposopathy, brown adipose tissue, ANGPTL3, ANGPTL4, ANGPTL8

## Abstract

Angiopoietin-like proteins, namely ANGPTL3-4-8, are known as regulators of lipid metabolism. However, recent evidence points towards their involvement in the regulation of adipose tissue function. Alteration of adipose tissue functions (also called adiposopathy) is considered the main inducer of metabolic syndrome (MS) and its related complications. In this review, we intended to analyze available evidence derived from experimental and human investigations highlighting the contribution of ANGPTLs in the regulation of adipocyte metabolism, as well as their potential role in common cardiometabolic alterations associated with adiposopathy. We finally propose a model of ANGPTLs-based adipose tissue dysfunction, possibly linking abnormalities in the angiopoietins to the induction of adiposopathy and its related disorders.

## 1. Introduction

Adipose tissue (AT) is an important metabolic organ and accounts for up to 25% of healthy individuals’ weight. Its main physiological role is to store energy as triglycerides-depots [1,2].

Unhealthy adipose tissue expansion is mainly due to fat accumulation in the abdominal and visceral compartments and usually associates with inflammation and metabolic abnormalities (e.g., insulin resistance, IR) [3]. Adiposopathy is the term used to describe these adverse anatomical and pathophysiologic changes in adipose tissue [4]. Adiposopathy is thought to be the main determinant of cardiometabolic disturbances such as type 2 diabetes mellitus (T2DM), non-alcoholic fatty liver disease (NAFLD), and thereby, atherosclerotic cardiovascular disease (ASCVD) [1,5,6].

The interplay of angiopoietin-like (ANGPTL)3, ANGPTL4, and ANGPTL8 is proved to regulate fatty acids trafficking during feeding and fasting through selective inhibition of lipoprotein lipase (LPL) in different metabolic compartments [7,8,9]. However, the angiopoietin-like system (ANGPTLs) metabolic function probably goes beyond this effect, and the angiopoietin-like system (ANGPTLs) is being recognized as a crucial regulator of adipose tissue functions [10,11,12].

In this review, we summarized the main features of adipose tissue subtypes, discussing the interplay between the ANGPTLs and adipocytes, and how this may influence cardiometabolic abnormalities.

## 2. Adipose Tissue Subtypes and Their Main Functions

Adipose tissue encompasses for three tissues that differ in the development, anatomical location, and metabolism: the white adipose tissue (WAT), the brown adipose tissue (BAT), and the beige adipose tissue.

### 2.1. WAT

WAT is most abundant in humans, and it may be distinguished into subcutaneous adipose tissue (SAT) and visceral adipose tissue (VAT) [13,14]. Despite the fact that WAT has been considered, for a long time, as a mere fat-storing organ, recent secretome studies have found that it is able to secrete endocrine and paracrine factors, allowing to define it as an active endocrine organ [15]. It is composed of mature and developing adipocytes, as well as fibroblasts, endothelial cells, and immune cells, namely, macrophages, neutrophils, eosinophils, mast cells, and T and B cells [16]. Given the abundance of resident inflammatory cells, recent evidence has also highlighted its role as a pro-inflammatory tissue [17,18].

The most important WAT-secreted mediators are adipokines, adiponectin, and leptin in particular. However, WAT also produces a whole set of cytokines, such as TNFα, MCP-1, IL-6, IL-10 [15,19]. All of these mediators express their actions both systemically, by affecting appetite, energy intake, lipid and glucose metabolism, and locally, by controlling macrophage infiltration and WAT inflammation. Moreover, adipokines secreted by omental and mesenteric VAT directly affect the liver, thus explaining their central role in regulating lipid and glucose metabolism [13,15,17,18].

Adiponectin is implicated in the metabolism of glucose and fatty acids. It has also a prominent role in improving insulin sensitivity, exerting anti-atherogenic actions, controlling metabolic homeostasis and modulating immune system function [20,21,22,23]. Unlike other adipocyte-derived hormones, adiponectin gene expression and blood concentrations are inversely associated with body mass index. Notably, this hormone also has a great effect on macrophages as it drives macrophage polarization through M2 (alternatively activated macrophage), inhibits M1 polarization (classically activated macrophages), thus reducing pro-inflammatory cytokine production and inducing immune-tolerance and an “immune-protective phenotype” [22,23].

Leptin is a pro-inflammatory mediator and unlike adiponectin, circulating levels of this hormone are proportional to individuals’ fat mass. Leptin centrally regulates body weight by linking nutritional status and neuroendocrine function [19,24]. Leptin also affects inflammation and immune responses [19,25]. Studies in leptin-deficient *ob/ob* mice showed that it is able to enhance T-cell survival and promotes T_H1_ polarization, which, in turn, stimulates a pro-inflammatory environment [24,26].

### 2.2. BAT

Unlike WAT, BAT is less abundant in humans. It can be found in the epicardium and near major blood vessels and accounts for 1–2% of adipose tissue’s total weight [13,27].

Most published knowledge on BAT derives from murine models and might not be fully applicable to humans. Nevertheless, studies in mouse models have highlighted that the main function of BAT is the non-shivering thermogenesis [28,29]. In fact, it is capable of rapidly producing large quantities of heat through the activation of uncoupling protein 1 (UCP1) located in the mitochondria inner membrane. BAT is particularly rich in β-adrenergic receptors (β-AR) and its function is activated by cold exposure via β_3_-AR activation by circulating catecholamines [13,30]. β_3_-AR stimulation enhances β-oxidation of triglycerides and induces upregulation of peroxisome proliferator-activated receptor γ coactivator-1α (PGC-1α), which, in turn, stimulates the expression of UCP1 and mitochondrial genes, thus leading to energy dissipation and thermogenesis [1,28,30,31].

In addition, BAT secretes several factors with paracrine function, named batokines, such as FGF21, VEGF, irisin, Slit2, as well as cytokines, such as IL-6, which are all crucial mediators of the crosstalk with WAT and skeletal muscle [1,32,33,34]. Batokines may greatly affect whole-body metabolic functions. In mouse models, BAT transplant led to increased catabolic rates, consistent weight reduction, and improvement in glucose tolerance, which could be attributed to paracrine and endocrine effects of the transplanted BAT on other tissues [1,33]. It must be noted that also the membrane content of cellular ceramides regulates BAT activity [35]. Mouse models deficient for serine palmitoyl-transferase long chain base subunit 2 (SPTLC2), a key enzyme in ceramides synthesis, showed increased metabolic activity in BAT and reduced WAT mass [35]. These effects are possibly mediated by the interaction of membrane ceramides with insulin downstream signaling pathway. Ceramides are membrane lipids capable of impairing insulin signaling through activation of protein kinase B (PKB), also known as AKT, which induces insulin receptor substrate 1-2 (IRS1-2) phosphorylation. Serine phosphorylated IRS proteins go through proteasome degradation, thereby increasing insulin resistance [35]. Similar to what was observed in animal models, in humans BAT seems to have a great impact on energy balance, despite its relatively small representation. Indeed, it plays a central role in lipid oxidation as its activation enhances the uptake of circulating free fatty acids (FFA) and WAT deposits’ utilization [28,36].

Schlessinger et al. found a similar gene expression and phenotype in the browning of murine and human WAT [37]. In this study, through a whole-transcriptome analysis, the authors found a subset of 49 genes that were commonly regulated or expressed in WAT browning in three different species (mouse, rhesus monkey, and human) [37]. Both non-human primates and mice are suitable models to study thermogenesis and insulin resistance in adipose tissue [37]. Another study investigated whether human subcutaneous white adipose tissue can adopt a BAT-like phenotype using a clinical model of prolonged and severe adrenergic stress [38]. Multilocular UCP1-positive adipocytes were found in subcutaneous WAT samples from burned subjects. UCP1 mRNA, mitochondrial density, and leak respiratory capacity in WAT increased after burn trauma. These data confirmed that in humans, WAT can shift from an energy-storing to an energy-dissipating tissue, in a process called browning [38].

Although most studies on batokines and their endocrine functions are based on mouse models, a specific BAT secretome was also found in cultured human adipocytes [39]. Deshmukh et al., by applying high-sensitivity mass-spectrometry-based proteomic in human adipocytes, found that a total of 101 proteins was exclusively quantified in brown adipocytes, and among these, ependymin-related protein 1 (EPDR1) has, apparently, a relevant role in the thermogenic determination during adipogenesis [39]. Interestingly, this study also found that, among the most significative *batokines* found in mice, only VEGF could be identified as secreted by human brown adipocytes. However, as limitations to this study, it must be recognized that weakly expressed proteins might not have been detected [39]. Nevertheless, the existence of a BAT-specific secretome in humans might be influential in energy metabolism regulation [39].

### 2.3. Beige Adipose Tissue

Beige adipose tissue is the latest discovered form of adipose tissue. It was first described in mice and it is considered to derive from trans-differentiation of WAT in the process named browning. [37,38,40]. Beige adipocytes are thought to be associated with the maintenance of energy balance, and this function may be similar to that of typical brown adipocytes [40].

Browning of adipose tissue is induced by intense adrenergic stimulation of WAT such as exposure to cold temperatures, intense exercise, bariatric surgery, and cancer cachexia [31,38]. All these signals can activate the PR domain containing 16 (PDRM16), a transcription factor [30,41], which in turn activates different regulating factors such as peroxisome proliferator-activated receptor γ coactivators (PGC-1α and PGC-1β) and peroxisome proliferator-activated receptor γ (PPARγ). The final effect is the induction of BAT-like adipogenesis in WAT precursors [30]. In humans, the formation of brown-like adipocytes within WAT, named beige adipocytes, has recently attracted much attention as a possible therapeutic target due to its inducible effect of dissipating extra-energy as heat [40].

## 3. Adipose Tissue Dysfunctions

The quality of adipose tissue depends on nutritional status and food intake. Excessive food intake may lead to WAT dysfunction, which is the typical abnormality found in metabolic syndrome [4,19]. As previously mentioned, the core function of WAT is to store lipids, primarily triglycerides. This needs rapid tissue remodeling in different metabolic conditions, thus involving the coordinated response of the different cellular components of WAT. Cell expansion was studied in different obese mouse models, showing that after 28 days of high-fat diet, a 4–6 fold increase in adipocyte volume is observed [42]. Reduced oxygen diffusion limits adipocyte volume expansion [1,42]. In conditions of mild hypoxia, a stress signal is generated in order to induce angiogenesis and extracellular matrix (ECM) remodeling to allow further adipose tissue expansion [42,43,44].

Conversely, in obese people, adipose depots expand beyond the tissue’s capacity of adequate angiogenesis, possibly resulting in persistent hypoxia, fibrosis, and adipocyte death [1,43,44]. However, studies on human adipose tissue oxygenation in dysfunctional conditions are still controversial: while some authors found SAT to be hypoxic in obese patients [43,44], others found it to be hyperoxic [45,46].

WAT becomes dysfunctional when fat-storing capacity is saturated, so that supraphysiologic conditions are necessary [43,45]. Individuals with WAT dysfunction characteristically have an imbalance in their adipokine profile and, typically, show reduced insulin sensitivity [42,47,48].

Brown adipose tissue may also become dysfunctional. BAT function declines with obesity and aging, giving it a “whitened” appearance, but the mechanisms contributing to this decline have been less defined. Firstly described by Shimizu et al., its existence has been exclusively reported in murine models [49]. However, it is noteworthy that the impairment of glucose uptake has been observed in BAT of T2DM patients [50]. Evidence is weak, but this tissue may be analog to murine whitened BAT [50]. In this condition, lipids accumulate in multiple lipid droplets (that is typical of BAT) becoming progressively a unilocular lipid deposit, endoplasmic reticulum goes through engulfment and UCP1 expression reduces [51]. Surprisingly, the whitened BAT starts to express leptin, which is a typical WAT marker [51,52]. The BAT whitening is possible due to the disruption of the “thermogenic signaling” and it is in particular attributed to increasing IR, disruption of β_3_-adrenergic signaling, and leptin resistance [51].

The phenomena of WAT beiging and BAT whitening are the clear demonstration that adipose tissue is an extremely flexible organ, capable to adapt to extreme supraphysiological conditions.

## 4. ANGPTLs

The angiopoietin-like (ANGPTL) protein system comprises different circulating factors showing similarity to angiopoietins. Eight different proteins are found in humans (ANGPTL 1–8) [7,8,9].

ANGPTL3, ANGPTL4, and ANGPTL8 are essential regulators of triglyceride and energy metabolism. In this paragraph, we will focus on their regulatory function on adipose tissue metabolism and dysfunction [10,11,12].

### 4.1. Role of the ANGPTL3-4-8 System in Regulating Triglycerides (TGs) Trafficking

The interplay of ANGPTL3, ANGPTL4, and ANGPTL8 is known as the crucial regulator of triglycerides (TGs) trafficking during feeding and fasting [7,53,54]. These ANGPTLs mainly act as inhibitors of lipoprotein lipase (LPL) and endothelial lipase (EL). These are the major extracellular enzymes that hydrolyze TGs carried by chylomicrons and very low-density lipoproteins (VLDLs), thus enabling FFA cellular replenishment and blood-TGs clearance [10,11,12].

ANGPTL3 is produced almost exclusively by the liver and acts as an hepatokine involved in the regulation of lipid and glucose metabolism [55,56,57,58]. It has been proposed that ANGPTL3 plays a role in the trafficking of energy substrates to either storage- or oxidative-tissues in response to food intake. It has a plasma TG-increasing effect, which is achieved by suppressing plasma TG clearance via inhibition of LPL activity [59]. The protein contains two functional domains: coiled-coil domain (N-terminus) and fibrinogen-like domain (C-terminus). Once secreted it is cleaved at the site located between residues 221–224 by proprotein convertases (PCSKs), mainly PCSK6 [59]. The cleaved N-terminal domain is sufficient and necessary for LPL inhibition [59]. The regulatory action of ANGPTL3 on TG breakdown is principally done during feeding, throughout its inhibitory effect on LPL in oxidative tissues such as BAT, heart, and skeletal muscles [7,60,61]. As a consequence, ANGPTL3 deficiency leads to an important reduction in circulating lipids, namely cholesterol, TGs, and HDL-C as described in Sun KK, a mouse KO-ANGPTL3 model, as well as in subjects affected by familial hypobetalipoproteinemia type 2 (FHBL2) [OMIM #60519], the human model of ANGPTL3 deficiency [54,57,59,62,63,64].

ANGPTL8 is mainly produced by the liver, but also by gut epithelium and adipose tissue [65,66]. While liver-derived ANGPTL8 is largely secreted in plasma, produced by adipocytes, remains intracellularly, particularly localized in nucleoplasm [67,68,69]. It is a small protein of 198 residues, capable of interaction with ANGPTL3, but exposing a coiled-coil domain only [7,70]. It is induced by feeding and facilitates the LPL inhibition activity by forming oligomers with ANGPTL3 [7,70]. It was recently proved that insulin acts as a powerful inducer of ANGPTL8 via PI3K/AKT signaling [68,71,72], and insulin-dependent ANGPTL8 induction occurs similarly in the liver and adipose tissue [68]. ANGPTL8 is also able to influence intracellular insulin signaling inducing AKT phosphorylation through a molecular pathway that is still unknown [73]. In summary, current evidence suggests that feeding induces insulin secretion, which in turn enhances ANGPTL8 expression in liver and adipose tissue. Then, ANGPTL8 accumulation upregulates insulin signaling, possibly in a feedforward fashion [73].

ANGPTL4 is produced by different cells throughout the body, mainly the liver, and adipose tissue [65,74]. It has a similar structure to ANGPTL3, it is induced by fasting and similarly acts as a LPL inhibitor, especially in WAT [75]. In comparison with ANGPTL3, ANGPTL4 is more active in oligomeric form, and cleavage from PCSKs does not hamper its LPL-inhibiting capacity [76]. Similar to ANGPTL3, also ANGPTL4 catalyzes LPL inhibition. Studies based on heparinized blood found that only homodimeric/multimeric LPL form is active [77]. Sukonina et al. [11], observed that treating LPL with ANGPTL4 inactivates LPL catalytic activity and increases the low-salt peak by heparin-sepharose chromatography, concluding that ANGPTL4 functions by converting active LPL homodimers into inactive monomers. However, more recently, Beigneux et al. [78] found LPL to be also active as a monomer, therefore ANGPTLs mediated LPL inhibition may have a different dynamic; while Mysling et al. [79] observed that ANGPTL4 inhibits LPL function by unfolding the hydrolase domain. ANGPTL4 was also found to promote intracellular cleavage of LPL mediated by PCSKs in adipose tissue of murine models [80]. ANGPTL3 and ANGPTL8 might have similar inhibition patterns.

ANGPTL4 is mainly induced by glucocorticoids, cold, and hypoxia-inducible factor 1α (HIF-1α). In particular, cold exposure induces ANGPTL4 differently in WAT (increased) and BAT (decreased), thus directing TGs towards BAT [81,82,83,84]. Proteomic studies found ANGPTL4 to be also expressed in the nucleoplasm in different cell lines [85,86]. However, its possible role in the nucleoplasm is still unknown, and its cellular localization should be further confirmed.

Chen et al. [70], found that ANGPTL8 is able to form complexes with both ANGPTL3 (in a 3:1 molar ratio) and ANGPTL4 (in 1:1 molar ratio) [70]. ANGPTL3/8 complexes are 100-times more potent in LPL inhibition than ANGPTL4/8 complexes, thus establishing a possible competitive mechanism in regulating LPL-activity in different tissues [70]

### 4.2. Role of the ANGPTL3-4-8 System in Regulating Adipose Tissue Function

ANGPTLs functions go probably beyond their effect in regulating lipid metabolism as they might be involved in the balance of whole-body energy through the regulation of adipose tissue physiology and substrates trafficking [54]. However, published evidence on ANGPTLs function beyond intracellular and extracellular lipolysis regulation is still fragmented. Most studies are based on murine models and results may not apply to humans.

Banfi et al. [61] found that knockout mouse models for both ANGPTL3 and ANGPTL8 had intense adipose tissue beiging in comparison with their wild type littermates. In particular, WAT of double KO mice expressed higher levels of UCP1, a typical BAT marker [61]. KO mice were also hypermetabolic, raising their basal temperature of 1 °C during feeding [61]. A similar effect could be evoked during fasting by the administration of a β_3_-agonist, or suppressed with a β_3_-antagonist during feeding [61]. It is still unknown how ANGPTL3 and ANGPTL8 may regulate the expression of UCP1. However, modification in levels of circulating lipids is possibly involved in enhancing β_3_-adrenergic receptor functioning, which is the main player in steering WAT *beiging* [30,31,61,87].

ANGPTL8 may be crucial in adipocyte differentiation and lipolysis regulation [67]. 3T3-L1 cells knocked down for ANGPTL8 show a consistent rise in mRNA expression for (PCG-1α) and UCP1 indicating differentiation towards BAT [67]. Moreover, ANGPTL4 levels and PPARs transcription factors were also raised indicating a shift towards lipolytic phenotype [67]. ANGPTL8 also has a core function in regulating chronic inflammation, in particular acting on TNF and IL-1, which are crucial mediators of adipose tissue metainflammation and lipolysis [88]. Zhang et al. [88] found that not only ANGPTL8 is induced by TNF and IL-1 signal, but it also facilitates the interaction of IKKγ (activated by NF-κB) with p62, thus labeling IKKγ for degradation and therefore, acting as an inflammation quencher [88]. ANGPTL8 treatment also reduces pro-inflammatory mediators, adipocyte survival, and replenishment [71]. Obese mouse models hyper-expressing ANGPTL8 show mitigation of adipose tissue inflammation and increased M2 polarization, probably due to the increased levels of secreted adiponectin [71]. Conversely, ANGPTL4 depletion in adipocytes results in better metabolic functions, since ANGPTL4 appears to stimulate the expression of SPTLC2, a core enzyme in ceramide production. Ceramides activate lipolysis through protein kinase C, zeta (PKCζ) (a Ca^2+^-dependent protein kinase) and induce proinflammatory mediators [35,89].

Since ANGPTLs seem to have a tight link with adipose tissue physiologic functions, it is possible that a disruption in ANGPTLs homeostasis may lead to adipose tissue dysfunction [54,90].

During feeding, serum glucose and insulin secretion enhance ANGPTL8 expression, in both adipocytes and hepatocytes [91]. A specific transcription factor for ANGPTL8 is the carbohydrate-responsive element-binding protein (ChREBP), which is induced by insulin stimulation [91]. While hepatocyte-derived ANGPTL8 is mainly secreted, in WAT and BAT it remains in nucleoplasm [69]. According to this model, ANGPTL8 is secreted in complexes with ANGPTL3, thus inhibiting LPL in oxidative tissues and directing dietary TGs to adipose tissue for storage [7,70]. Enhanced ANGPTL8 is associated with reduced activation of AMPK and PPARα inhibition (that regulates ANGPTL4 expression) [92,93]. Comprehensively, ANGPTL8 expression in feeding ameliorates insulin sensitivity and blocks lipolysis, thus preparing adipocytes to receive nutrients [67] (Figure 1).

During fasting, ANGPTL4 expression is upregulated in both hepatocyte and adipocytes through activation of PPARs transcription factor (PPARα and PPARγ in particular) in response to glucagon and glucocorticoids [93,94,95,96]. In the liver, ANGPTL4 is proved to be secreted in fasting and during exercise, probably carrying out endocrine functions [94]. Increased ANGPTL4 expression is associated with increased gluconeogenesis and lipogenesis, which lead to VLDL secretion and fatty-liver in mouse models [89]. In adipose tissue, ANGPTL4 is secreted as a paracrine factor, since it remains bound to ECM surrounding adipocytes [75,80]. Therefore, PCSKs secretion in response to fasting might be crucial in liberating the N-terminus domain of ANGPTL4, thus enforcing adipose tissue LPL inhibition and directing TGs flux to BAT and muscular tissue for catabolism [89,97] (Figure 1).

In conditions of chronic overnutrition, both serum insulin and glucocorticoid levels increase, leading to a change in ANGPTL system homeostasis: chronic overexpression of ANGPTL8 in hepatocytes may determine augmented insulin resistance, lipogenesis, increased VLDL secretion, and liver steatosis [68,91]. In WAT, the co-expression of ANGPTL4 and ANGPTL8 may determine enhanced adipose tissue inflammation and uncontrolled expansion leading to adipocytes dysfunction [35,88,91]. In this context, BAT may be also affected, since a chronic overexpression of ANGPTL8 reduces PPARs and UCP1 expression favoring lipogenesis and BAT whitening [51] (Figure 2).

## 5. Clinical Conditions Linking ANGPTLs and Adipose Tissue Dysfunction

To further highlight the role of ANGPTLs as mediators of unhealthy adipose tissue expansion, it may be relevant to examine changes in the ANGPTL3-4-8 system in pathological conditions in which adipocytes play a pivotal role, namely T2DM/insulin-resistant state, NAFLD/NASH, and lipodystrophies (Table 1).

### 5.1. Type 2 Diabetes Mellitus and Insulin Resistant State

Type 2 Diabetes Mellitus (T2DM) and state of insulin resistance, seem to be tightly associated with dysregulation of angiopoietin-like 3-4-8 system.

ANGPTL3 and ANGPTL4 levels are increased in T2DM when compared with obese non-diabetic patients and healthy controls [99,100]. A strong correlation was seen among circulating ANGPTL3 and ANGPTL8 indicating a possible co-secretion of the two proteins [99,100]. Recently, a phase II trial has tested vupanorsen [101], an ANGPTL3 antisense oligonucleotide, in T2DM patients [101]. Preliminary results of the study found no significant change in the HOMA index, a widely used index for insulin resistance [102]. Instead, a significant reduction in circulating TGs and non-HDL cholesterol was found [102].

Cinkajzlová et al. [103] evaluated the levels of ANGPTL3 and ANGPTL4 in T2DM compared with those observed in extreme nutritional states (anorexia nervosa, short bowel syndrome, and extreme obesity) [103]. Despite the small number of investigated subjects, ANGPTL3 and ANGPTL4 were found to be strictly associated with energy intake and previous metabolic status. Bariatric surgery reduces ANGPTL3 levels only for a short period in obese patients, and ANGPTL3 levels were back at pre-surgery levels within 12 months [103]. On the contrary, in patients with anorexia nervosa re-alimentation reduces ANGPTL3 levels. In the same study, ANGPTL4 levels moved in the opposite direction in T2DM patients, by increasing in fasting state [103]. A strong negative correlation between levels of circulating ANGPTL4 and both BMI and HOMA-index were observed in obese children [102,104]. However, ANGPTLs serum quantification has no international validated standard, and high-grade variability is reported among different publications. Further investigations are needed to assess the reproducibility of these data.

Levels of circulating ANGPTL8 are apparently crucial actors in impairing glucose metabolism but results are conflicting [61,73,105,106]. In most human studies, serum ANGPTL8 is measured through commercial ELISA kits that showed high inter-, and intra-assay variability. Reported evidence might not be reliable. Morinaga et al. using a self-developed and controlled ANGPTL8 assay found higher circulating levels of ANGPTL8, in T2DM patients, associated with increased levels of LDL-C, TGs, and reduced HDL-C [101]. Interestingly, differences in ANGPTL8 expression are observable in obese vs. non-obese patients: in obese patients, VAT shows increased levels of ANGPTL8 in the nucleoplasm, while serum levels remain similar to non-obese controls [69].

Although in literature alterations in circulating levels of the ANGPTLs in T2DM are still uncertain, they seem to exert an important role. Since insulin resistance positively relates to circulating TGs, inhibition of one component of the ANGPTLs system might determine an increase in insulin sensitivity [107]. However, further studies are needed to clarify the role of ANGPTLs in T2DM and its treatment potentials.

### 5.2. Non-Alcoholic Fatty Liver Disease and Non-Alcoholic Steatohepatitis

The liver is the main regulator of glucidic and lipidic metabolism and hepatocytes produce whole body circulating ANGPTL3, ANGPTL8, and ANGPTL4 [7,65].

Levels of circulating ANGPTL3 are significantly increased in NAFLD/NASH and this is associated with an impairment of insulin sensitivity [108]. The same results are found in advanced liver steatosis or steatohepatitis, where an increase in circulating levels of ANGPTL3 suggests a relation between liver inflammation and ANGPTL3 secretion [100].

In vitro observations have highlighted that ANGPTL3 has probably a role in regulating hepatocyte lipid metabolism [64]. HepG2 cells knocked out for ANGPTL3, showed important changes in lipid metabolism as outlined by lipidomic studies [62,64]. ANGPTL3 KO in liver-derived cell lines, caused intracellular ApoB accumulation [64] and a change in intracellular lipidic content, towards longer and more unsaturated fatty acid chain as poly-unsaturated fatty acid (PUFA), which may be protective towards NAFLD and NASH [62,64].

These in vitro results have been partially confirmed in humans. Indeed, Differently from FHBL1 (Familial hypobetalipoproteinemia OMIM #615558), FHBL2 patients show a lower prevalence of ultrasound-determined liver steatosis than the general population, and no evidence of increased markers of hepatocyte death or bilio-stasis [109,110].

Preliminary results on hypertriglyceridemic patients treated with vupanorsen [101], an antisense ANGPTL3 inhibitor, did not show any significant differences in hepatic fat fraction, although levels of serum triglycerides dropped significantly [101]. The associations between ANGPTL3, lipoprotein metabolism and liver health status have been analyzed in the DiOGenes (diet, obesity, and genes) study [111]. Levels of ANGPTL3 were measured before and after an 8-week period of a low-calorie diet in obese patients suffering from NAFLD [111]. The major findings were a strong negative correlation between plasma ANGPTL3 concentration and AST levels, and a positive association with cytokeratin 18 (CK-18). Plasma CK-18 acts as a biomarker for the apoptotic death of hepatocytes, NASH, and hepatic inflammation [111]. Therefore, a rise in circulating ANGPTL3 in NAFLD/NASH patients might be determined by increased liver inflammation and hepatocyte death [111].

Circulating ANGPTL8 has been positively related to levels of lipid accumulation in hepatocytes [68,92]. Leptin deficient *ob/ob* mice treated with high fat diet show increased levels of circulating ANGPTL8 which is directly related to the grade of liver steatosis [112].

ANGPTL4 may also be involved in NAFLD. Indeed, increased levels of circulating ANGPTL4 induce both lipogenesis and activation of PKCζ and ceramides production, leading to a pro-inflammatory microenvironment and facilitating progression in NAFLD and NASH [89].

Although ANGPTLs seem to exert an important role in the coordinate regulating of liver and adipose tissue functions, the intracellular mechanisms underlying the regulation of hepatic lipid metabolism mediated by ANGPTLs is still unknown and should be further investigated.

### 5.3. Lipodystrophy

Lipodystrophy is a rare disorder characterized by hypoleptinemia and partial or complete absence of adipose tissue; it is associated with insulin resistance and hypertriglyceridemia [113,114]. Circulating levels and hepatic expression of ANGPTL3 are increased in leptin-resistant *db/db* mice and leptin-deficient *ob*/*ob* mice [114]. Administration of leptin to *ob*/*ob* mice decreases hepatic ANGPTL3 mRNA expression and plasma ANGPTL3 levels [114].

Recently, Muniyappa et al. [115] found that patients affected by lipodystrophy treated with metreleptin showed a marked reduction in circulating ANGPTL3 levels together with a consistent improvement in liver steatosis, whereas ANGPTL4 levels remained unchanged [115].

These results highlight a possible role of ANGPTL3 in the regulation of adipose tissue replenishment and liver-adipose tissue lipid exchange mediated by leptin.

## 6. Conclusions

Adipose tissue represents the regulatory center of energy expenditure and intake. Intense crosstalk with the liver and muscles is crucial to fulfilling body lipid necessities for energy supplies and membrane construction purposes.

The angiopoietin-like protein system is emerging as a regulator of whole-body energy metabolism, capable of regulating key adipose tissue functions, and possibly leading to adipose tissue dysfunction. Excessive food intake impairs the whole system, which has evolved to store the maximal amount of energy, leading to adipose tissue dysfunction, ANGPTLs hypersecretion, and the development of cardiometabolic diseases.

ANGPTL-3, -4, and -8 are interesting pharmacological targets for cardiometabolic conditions. It is clear that a rise in serum levels of one or more of them may be considered a marker for metabolic deregulation and cardiometabolic disease development. Further research will hopefully highlight newer and interesting aspects of intracellular ANGPTLs functions and lipid metabolism modification in patients treated with the new ANGPTLs lowering drugs.

## Figures and Tables

**Figure 1 ijms-22-00742-f001:**
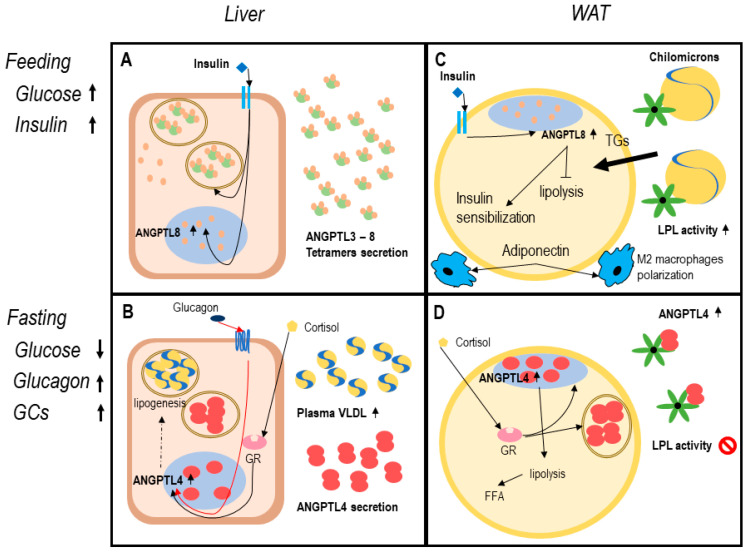
The angiopoietin-like system (ANGPTLs) model in healthy adipose tissue. Legend: functioning of ANGPTL system in healthy adipose tissue. Dotted lines report data with a low level of evidence. Panels (**A**,**B**) describe the hepatocyte function in feeding and fasting. In response to insulin increase (**A**), hepatocytes secrete ANGPTL8, and ANGPTL3. ANGPTL3-8 tetramers inhibit lipoprotein lipase (LPL) in oxidative tissues [7,68]. Reduced blood glucose in fasting (**B**) leads to increased circulating glucocorticoids and glucagon that induce ANGPTL4 expression [82,94]. In the liver, ANGPTL4 expression possibly enhances lipogenesis and gluconeogenesis, VLDL production, and secretion, as proved in mouse models [89]. Panels (**C**,**D**) describe white adipocyte function in feeding and fasting. During feeding (**C**), insulin enhances nuclear expression of ANGPTL8 determining a block in lipolysis and enhanced insulin sensibilization, adiponectin secretion, and M2 polarization of resident macrophages [67,68,71]. In fasting (**D**) ANGPTL4 is expressed both in nucleoplasm and secreted [74,75,84,86,94]. In this physiologic condition, lipolysis is functional to liberate free fatty acids (FFA) deposit and promotes energy utilization [7,81,89].

**Figure 2 ijms-22-00742-f002:**
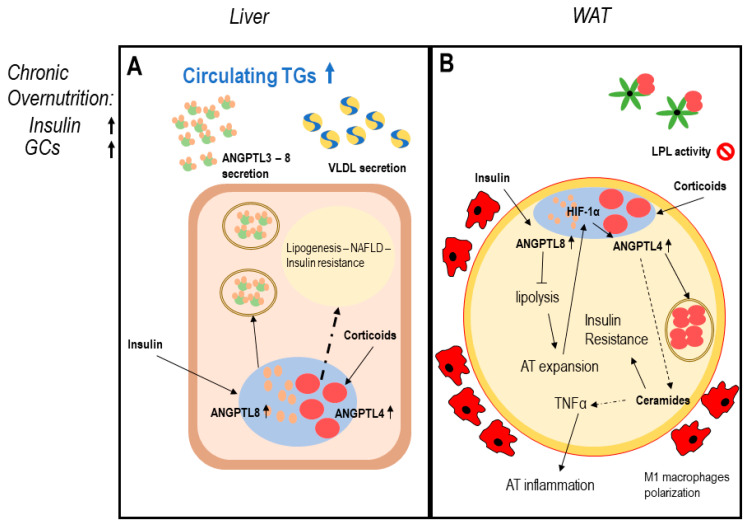
ANGPTLs model of adipose tissue dysfunction. Legend: possible induction mechanism of adipose tissue dysfunction involving ANGPTL3-4-8. Dotted lines report data with a low level of evidence. Panel (**A**) describes dysregulation of hepatocytes function in conditions of chronic overnutrition: overexpressed insulin levels together with elevated plasma glucocorticoids in obese patients induce both ANGPTL8 and ANGPTL4 expression in hepatocytes [68,73,74,75,84,86,89,94]. ANGPTL4 possibly induces ceramides production; therefore, worsening hepatic insulin resistance and liver steatosis [35,81,89]. Panel (**B**) describes white adipose tissue (WAT) dysfunction in conditions of chronic overnutrition, potentially mediated by ANGPTLs system disruption: excess insulin induces overexpression of nuclear ANGPTL8 in WAT [67,68]. ANGPTL8 in turn enhances insulin signaling via AKT phosphorylation, chronic AKT phosphorylation induces worsening insulin resistance [73]. Permanent block in lipolysis leads to adipocyte mass expansion over oxygen diffusion capacity, this creates adipocyte stress and hypoxia-inducible factor 1-alpha (HIF1α) induction, which in turn induces ANGPTL4 expression [83,84], together with enhanced circulating glucocorticoids, typical of overnutrition. ANGPTL4 partly blocks WAT LPL, partly induces lipolysis and ceramide production [89]. Ceramide production is associated with worsening insulin resistance (IR) and inflammation, leading to WAT secretion of proinflammatory mediators, M1 macrophage infiltration, and adipocyte death typical of AT dysfunction [35,98].

**Table 1 ijms-22-00742-t001:** Serum ANGPTLs change in metabolism-related diseases.

	T2DM	Obesity	NASH/NAFLD	Lipodystrophy
ANGPTL3	↑ in T2DM vs. obese non-T2DM. ^b^	Unchanged ^b^	Non-increased in NAFLD, higher levels in advanced levels of NASH. ^b^ ↑ in conditions of metabolic inflammation. ^b^ ↓ in lipodystrophic patients in treatment with metreleptin. ^b^	High levels of ANGPTL3 in untreated patients. ^b^ Consistent ANGPTL3 reduction in leptin-replacement therapy. ^b^
ANGPTL4	↑ in T2DM vs. obese non-T2DM. ^b^	↑ Obesity vs. normal BMI. ^b^ Reflects adipose tissue dysfunction. ^b^	Unchanged ^b^	Unchanged ^b^
ANGPTL8	↑↑ in glycometabolism impairment. ^b^ Induced by insulin. ^a^ Related with serum LDL-c, TGs and HDL-c inversely. ^b^	Unchanged in case of glycometabolism compensation. ^b^	Associated with grade of lipid accumulation in hepatocytes. ^a^	Associated with grade of lipid accumulation in hepatocytes in leptin deficient mouse. ^a^ Unresearched in Human. ^b^

Table Legend: Levels of circulating ANGPTLs in common cardiometabolic diseases. ^a^—studies in animal models. ^b^—studies in human. Arrows indicate increase (↑) or decrease (↓) of ANGPTLs serum concentration. T2DM (type 2 diabetes mellitus); NAFLD (Non-Alcoholic fatty liver Disease); NASH (Non-Alcoholic Steatohepatitis); BMI (body mass index); TGs (triglycerides); HDL-c (high density lipoprotein cholesterol); LDL-c (low density lipoprotein cholesterol).

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
