# Peer review of "The Interplay between Angiopoietin-Like Proteins and Adipose Tissue: Another Piece of the Relationship between Adiposopathy and Cardiometabolic Diseases?"

_ijms, 2021, doi:10.3390/ijms22020742_

Round 1

Reviewer 1 Report

This manuscript reviews the literature on ANGPTLs, with a strong focus on adipose tissue. Given the ongoing interest in ANGPTLs as therapeutic targets, the review is timely and could be a valuable addition to the literature after incorporating substantial changes. I have three major concerns and a number of specific corrections.

1) In general I would encourage the authors to be more critical towards the published literature. We cannot automatically assume that all published data are correct. Certain findings are solid because they have been reproduced many times. Other findings may be correct but still need confirmation. Other findings are simply wrong because they used the wrong tools (for instance, ELISAs that don’t work). This needs to be taken into account when writing a review.

2) The paper has a tendency to cite other reviews (especially reference 48) instead of the primary literature. This is undesirable. Please only cite the primary literature and make sure the primary literature is properly described. See line 216, 223, 226, 269, 272. Other reviews may make inferences that do not appropriately reflect the primary literature.

3) The paper mainly focuses on mouse data, without taking into consideration if these data apply to humans as well, especially in the first part of the paper (sections 1-3).

Other comments

1) The review requires English language editing. The use of grammar software such as Grammarly is recommended, although this will probably not fix all corrections needed.

2) Line 55. Please remove Resistin. There is no evidence that Resistin is produced and secreted by human adipose tissue. Accordingly, lines 75-82 should be removed. People keep referring to resistin as adipokine when it really isn’t.

3) What the purpose of including a table of adipokines in a paper on ANGPTLs? This is distracting and out of scope. I would suggest removing the table and downsizing sections 1-3 of the paper.

4) The section on BAT mixes information that is generic and information that is specific to mice or other animals (and is not applicable to human). This makes it rather confusing. It is important to clarify which findings are based on studies in mice and which findings are based on studies in human or have been validated in humans. Is there any evidence that human BAT produces batokines, including the ones mentioned (line 103).

5) Line 130: Is there any evidence for browning in human adipose tissue under relevant physiological conditions?

6) Top of page 4. There is no evidence that obesity is associated with adipose tissue hypoxia in humans. To the contrary, it seems that oxygen pressure in adipose tissue is higher in obese subjects.

7) Line 160. “However, it has been also shown that BAT goes through a “whitening” process in 160 patients affected by metabolic syndrome [32]. “ Reference 32 only describes studies performed in mice. What is the basis for this statement? Is there any evidence for BAT whitening in humans?

8) Line 215. It is stated that ANGPTL4 is produced by the pancreas but the reference provided doesn’t even once mention pancreas. Please remove pancreas.

9) The article mentions the cleavage of LPL multimers/dimers by ANGPTLs. However, recent studies by Steve Young and colleagues have suggested that LPL may not form dimers and that inhibition of LPL by ANGPTLs may not involved dimer dissociation. These notions should be mentioned.

10) Evidence supporting a role for ANGPTLs beyond extracellular and intracellular lipolysis is rather fragmented and still lacks consistency. This should be more clearly articulated in the paper.

11) Line 271. This statement is incorrect in many ways. ANGPTL4 is induced in liver by PPARa, not PPARg. Second, the effect of fasting on ANGPTL4 expression in the liver is independent of PPARa (PMID: 10862772). Third, the induction of ANGPTL4 by fasting in adipose tissue is not mediated by PPARg. The authors should refrain from citing other reviews, which propagates these mistakes (see main comments).

12) The statement that ANGPTL4 is not secreted from hepatocytes is incorrect. Again, the authors refer to a review. It has been shown that human ANGPTL4 can be detected in plasma of hepatocyte humanized mice. Also, fibrates raise plasma levels of ANGPTL4 in humans, supporting the idea that hepatocytes secrete ANGPTL4, which ends up in the circulations. Finally, liver-ANGPTL4 overexpression inhibits The prevailing notion is that liver-derived ANGPTL4 has an endocrine function, in contrast to adipose tissue-derived ANGPTL4.

13) The evidence supporting the notion that ANGPTL4 promotes lipogenesis is very weak, certainly not enough to justify figure 1B.

14) Figure 1D. Please remove part on ceramide and TNFa. This finding is still contentious and has no direct relevance for the role of ANGPTL4 in WAT during fasting.

15) Figure 1E and F. The role of ANGPTL4 in BAT should be shown in the context of cold exposure, not fasting.

16) Figure 1 implies that ANGPTL4 protein is produced in the nucleus. Please adjust.

17) Figure 2: please remove link between HIF1a and ANGPTL4. There is no evidence that human WAT is hypoxic during obesity and that hypoxia may influence ANGPTL4 expression during adipose tissue dysfunction in humans.

18) Section 5 needs to be rewritten taking into account that nearly all studies that report on ANGPTL8 in human plasma are wrong due to the use of the non-functional ELISA. These data simply cannot be trusted. Data on plasma ANGPTL4 and ANGPTL3 that used the R&D ELISA can be trusted.

Author Response

POINT BY POINT REPLY

We thank the reviewer for the careful evaluation of the manuscript, thoughtful considerations and suggestions. Our reply to the points they have raised is reported below. (reference to lines are applicable to clear form of revisioned manuscript)

Reviewer 1

This manuscript reviews the literature on ANGPTLs, with a strong focus on adipose tissue. Given the ongoing interest in ANGPTLs as therapeutic targets, the review is timely and could be a valuable addition to the literature after incorporating substantial changes.

We thank the reviewer for recognizing the importance of the topic.

 I have three major concerns and a number of specific corrections.

1) In general, I would encourage the authors to be more critical towards the published literature. We cannot automatically assume that all published data are correct. Certain findings are solid because they have been reproduced many times. Other findings may be correct but still need confirmation. Other findings are simply wrong because they used the wrong tools (for instance, ELISAs that don’t work). This needs to be taken into account when writing a review.

We agree with the reviewer that some data have not been replicated and still need to be confirmed in other studies but our intention was to summarize all available literature regarding ANGPTLs and adipose tissue. Moreover, although we are aware that some methods have wide variability when measuring plasma concentrations of ANGPTLs (i.e. plasma ANGPTL8 levels by ELISA), we believed that it might have been interesting to report the different results as they were published acknowledging their limitation.

Then, to comply with the reviewer’s suggestion we have revised the manuscript: reporting the published data with a more critical approach, considering those which appeared more solid and eliminating the weaker ones

2) The paper has a tendency to cite other reviews (especially reference 48) instead of the primary literature. This is undesirable. Please only cite the primary literature and make sure the primary literature is properly described. See line 216, 223, 226, 269, 272. Other reviews may make inferences that do not appropriately reflect the primary literature.

In the revised version of the manuscript, we have followed the reviewer’s indication by retrieving primary literature and limiting the number of other reviews cited. In addition, each time a review was cited as reference, primary literature was added. Therefore, the remaining review articles cited are Ref. #1, #7, #13, #16, #19, #24, #28, #45, #78, and #113. Nevertheless, we cited ref.# 78 (ex ref. #48) only referring to 3 lines: 223, 300, and 307. Primary literature has been always added.

3) The paper mainly focuses on mouse data, without taking into consideration if these data apply to humans as well, especially in the first part of the paper (sections 1-3).

We agree that mouse data not always correspond to observations in human. However, ANGPTLs are relatively new discovered proteins and also the reviewer has to admit that the majority of evidences on their function and their link to adipose tissue have been collected mainly using in vitro or mouse models studies. Therefore, we organized the paper firstly revising evidences obtained by basic science studies with the aim to better understand ANGPTLs functioning. Then, in the second part of the manuscript, we revised the evidences on the role of ANGPTLs in adipose tissue dysfunction and metabolic diseases in humans. Nevertheless, references to mouse data were always declared and we highlighted the limitation of these models throughout the manuscript. Changes in the manuscript have been made as follows:

  • Starting from line 78: “Most published knowledge on BAT derives from murine models and might not be fully applicable to humans. Nevertheless, studies in mouse models have highlighted that the main function of BAT is the non-shivering thermogenesis [28], [29].”

  • Starting from line 90: “Batokines may greatly affect whole-body metabolic functions. In mouse models, BAT transplant determined increased catabolic rates, consistent weight reduction, and improvement in glucose tolerance, which could be attributed to paracrine and endocrine effects of the transplanted BAT on other tissues [1], [33].”

  • Starting from line 101: “Similar to what was observed in animal models, in humans BAT seems to have a great impact on energy balance, despite its relatively small representation. Indeed, it plays a central role in lipid oxidation as its activation enhances the uptake of circulating free fatty acids (FFA) and WAT deposits’ utilization [28], [36].”

  • Starting from line 108: “Both non-human primates and mice are suitable models to study thermogenesis and insulin resistance in adipose tissue [37]. Another study investigated whether human subcutaneous white adipose tissue can adopt a BAT-like phenotype using a clinical model of prolonged and severe adrenergic stress [38].”.

  • Starting from line 116: “Although most studies on batokines and their endocrine functions are based on mouse models, a specific BAT secretome was also found in cultured human adipocytes [39]. Deshmukh et al., by applying high-sensitivity mass-spectrometry-based proteomic in human adipocytes, found that a total of 101 proteins was exclusively quantified in brown adipocytes, and among these, ependymin-related protein 1 (EPDR1) has apparently a relevant role in the thermogenic determination during adipogenesis [39]. Interestingly, this study also found that, among the most significative batokines found in mice, only VEGF could be identified as secreted by human brown adipocytes. However, as limitations to this study, it must be recognized that little expressed proteins might not have been detected [39]. Nevertheless, the existence of a BAT-specific secretome in humans might be influential in energy metabolism regulation [39].”.

  • Starting from line 146: “Cell expansion was studied in different obese mouse models, showing that after 28 days of high-fat diet, a 4-6 fold increase in adipocyte volume is observed [42]

  • Starting from line 151: “Conversely, in obese people adipose depots expand beyond the tissue’s capacity of adequate angiogenesis, thus resulting in persistent hypoxia, fibrosis, and adipocyte death [1], [43], [44].”

  • Starting from line 159: “Firstly described by Shimizu et al., its existence has been exclusively reported in murine models [47]. However, it is noteworthy that the impairment of glucose uptake has been observed in BAT of T2DM patients [48]. Evidence is weak, but this tissue may be analog to murine whitened BAT [48].”

  • Starting from line 230: “ANGPTLs functions go probably beyond their effect in regulating lipid metabolism as they might be involved in the balance of whole-body energy through the regulation of adipose tissue physiology and substrates trafficking [84]. However, published evidence on ANGPTLs function beyond intracellular and extracellular lipolysis regulation is still fragmented. Most studies are based on murine models and results may not apply to humans.

Other comments

1) The review requires English language editing. The use of grammar software such as Grammarly is recommended, although this will probably not fix all corrections needed.

We thank the reviewer for the suggestions. Accordingly, we have extensively revised the text by using also the recommended grammar software.

2) Line 55. Please remove Resistin. There is no evidence that Resistin is produced and secreted by human adipose tissue. Accordingly, lines 75-82 should be removed. People keep referring to resistin as adipokine when it really isn’t.

We apologize for this inaccuracy, as resistin should not be listed among adipokines. Therefore, we have removed resistin from the paragraph related to adipokines together with ref. #19-20.

3) What the purpose of including a table of adipokines in a paper on ANGPTLs? This is distracting and out of scope. I would suggest removing the table and downsizing sections 1-3 of the paper.

We agree with the reviewer that Table 1 can be eliminated. Changes in the text have been made accordingly.

4) The section on BAT mixes information that is generic and information that is specific to mice or other animals (and is not applicable to human). This makes it rather confusing. It is important to clarify which findings are based on studies in mice and which findings are based on studies in human or have been validated in humans. Is there any evidence that human BAT produces batokines, including the ones mentioned (line 103).

We acknowledge that this section might be confusing. Therefore, we tried to clarify the evidences on BAT further highlighting which ones are based on mouse models and which in humans. Then, substantial corrections were made as follows:

  • Starting from line 76: “Unlike WAT, BAT is less abundant in humans. It can be found in the epicardium and near major blood vessels and accounts for 1-2% of adipose tissue’s total weight [13], [27]. Most published knowledge on BAT derives from murine models and might not be fully applicable to humans. Nevertheless, studies in mouse models have highlighted that the main function of BAT is the non-shivering thermogenesis [28], [29].”

  • Starting from line 88: “In addition, BAT secretes several factors with paracrine function, named as batokines, such FGF21, VEGF, irisin, Slit2, as well as cytokines such IL-6, which are all crucial mediators of the crosstalk with WAT and skeletal muscle [1], [32]–[34]. Batokines may greatly affect whole-body metabolic functions. In mouse models, BAT transplant determined increased catabolic rates, consistent weight reduction, and improvement in glucose tolerance, which could be attributed to paracrine and endocrine effects of the transplanted BAT on other tissues [1], [33].

  • Starting from line 101: “Similar to what was observed in animal models, in humans BAT seems to have a great impact on energy balance, despite its relatively small representation. Indeed, it plays a central role in lipid oxidation as its activation enhances the uptake of circulating free fatty acids (FFA) and WAT deposits’ utilization [28], [36].

Schlessinger et al. found a similar gene expression and phenotype in the browning of murine and human WAT [37]. In this study, through a whole-transcriptome analysis, the authors found a subset of 49 genes that were commonly regulated or expressed in WAT browning in three different species (mouse, rhesus monkey, and human) [37]. Both non-human primates and mice are suitable models to study thermogenesis and insulin resistance in adipose tissue. [37] Another study investigated whether human subcutaneous white adipose tissue can adopt a BAT-like phenotype using a clinical model of prolonged and severe adrenergic stress [38]. Multilocular UCP1-positive adipocytes were found in subcutaneous WAT samples from burned subjects. UCP1 mRNA, mitochondrial density, and leak respiratory capacity in WAT increased after burn trauma. These data confirmed that in humans, WAT can shift from an energy-storing to an energy-dissipating tissue, in a process called browning [38].

Although most studies on batokines and their endocrine functions are based on mouse models, a specific BAT secretome was also found in cultured human adipocytes [39]. Deshmukh et al., by applying high-sensitivity mass-spectrometry-based proteomic in human adipocytes, found that a total of 101 proteins was exclusively quantified in brown adipocytes, and among these, ependymin-related protein 1 (EPDR1) has apparently a relevant role in the thermogenic determination during adipogenesis [39]. Interestingly, this study also found that, among the most significative batokines found in mice, only VEGF could be identified as secreted by human brown adipocytes. However, as limitations to this study, it must be recognized that little expressed proteins might not have been detected [39]. Nevertheless, the existence of a BAT-specific secretome in humans might be influential in energy metabolism regulation [39].”.

5) Line 130: Is there any evidence for browning in human adipose tissue under relevant physiological conditions?

Although not extensively reproduced, there is some evidence of adipose tissue browning in men. (PMID:26244931; 26308478). These data have been discussed in the revised version of manuscript at paragraph 2.2 and related references have been added.

6) Top of page 4. There is no evidence that obesity is associated with adipose tissue hypoxia in humans. To the contrary, it seems that oxygen pressure in adipose tissue is higher in obese subjects.

Although we are aware that data are not univocal, we found 2 independent studies by Lawler et al. (PMID: 26871994, Ref. 43) and Cifarelli et al. (PMID: 33164985, Ref 44) that found adipose tissue to be hypoxic in obese patients. In both studies, authors compared obese vs non-obese patients and studied insulin resistance. Adipose tissue oxygenation was measured through direct invasive catheters, further confirming the possibility that obesity is associated with adipose tissue hypoxia in humans. However, Authors concluded that hypoxia may be a simply a consequence of adipose tissue expansion and its relationship with pathogenesis of obesity-induced insulin resistance still remain to be clarified.

According with the consideration mentioned above, we decided not to make any changes.

7) Line 160. “However, it has been also shown that BAT goes through a “whitening” process in 160 patients affected by metabolic syndrome [32]. “Reference 32 only describes studies performed in mice. What is the basis for this statement? Is there any evidence for BAT whitening in humans?

We apologize for the mistake and the sentence has been deleted.

Following reviewer’s suggestion we revised the literature. We found little evidence of indirect proof of BAT whitening in men. Blondin et. al (10.2337/db14-1651, ref 38) found impaired glucose uptake and reduced density upon computed tomography (CT) in BAT of obese men concluding that cold-induced NEFA uptake and oxidative metabolism are not defective in type 2 diabetes despite reduced glucose uptake per BAT volume and BAT “whitening”. Therefore, the sentences at Page 4, line 159 have been changed as follows: “Firstly described by Shimizu et al., its existence has been exclusively reported in murine models [47]. However, it is noteworthy that the impairment of glucose uptake has been observed in BAT of T2DM patients [48]. Evidence is weak, but this tissue may be analog to murine whitened BAT [48].”

8) Line 215. It is stated that ANGPTL4 is produced by the pancreas but the reference provided doesn’t even once mention pancreas. Please remove pancreas.

We apologize for the wrong reference. The Human Protein Atlas project found ANGPTL4 to be produced by the pancreas and therefore has been added this information as references #63 and #72.

(https://www.proteinatlas.org/ENSG00000167772-ANGPTL4; doi: 10.1126/science.1260419)

9) The article mentions the cleavage of LPL multimers/dimers by ANGPTLs. However, recent studies by Steve Young and colleagues have suggested that LPL may not form dimers and that inhibition of LPL by ANGPTLs may not involve dimer dissociation. These notions should be mentioned.

We thank the reviewer for suggesting this interesting study. We decided to mention that LPL is also active as a monomer in Paragraph 4.1 at page 5, from line 213 as follows: “Similar to ANGPTL3, also ANGPTL4 catalyzes LPL inhibition. Studies based on heparinized blood found that only homodimeric/multimeric LPL form is active [75]. Sukonina et al. [11], observed that treating LPL with ANGPTL4 inactivates LPL catalytic activity and increases the low-salt peak by heparin-sepharose chromatography, concluding that ANGPTL4 functions by converting active LPL homodimers into inactive monomers. However, more recently Beigneux et al. [76], found LPL to be also active as a monomer, therefore ANGPTLs mediated LPL inhibition may have a different dynamic. Moreover, Mysling et al. [77], observed that ANGPTL4 inhibits LPL function by unfolding the hydrolase domain. ANGPTL3 and ANGPTL8 might have a similar inhibition pattern.”.

10) Evidence supporting a role for ANGPTLs beyond extracellular and intracellular lipolysis is rather fragmented and still lacks consistency. This should be more clearly articulated in the paper.

We agree with reviewer’s suggestion. We have therefore changed the sentences in paragraph 4.2 at page 5, from line 230 as follows: “ANGPTLs functions go probably beyond their effect in regulating lipid metabolism as they might be involved in the balance of whole-body energy through the regulation of adipose tissue physiology and substrates trafficking [84]. However, published evidence on ANGPTLs function beyond intracellular and extracellular lipolysis regulation is still fragmented. Most studies are based on murine models and results may not apply to humans.”.

11) Line 271. This statement is incorrect in many ways. ANGPTL4 is induced in liver by PPARa, not PPARg. Second, the effect of fasting on ANGPTL4 expression in the liver is independent of PPARa (PMID: 10862772). Third, the induction of ANGPTL4 by fasting in adipose tissue is not mediated by PPARg. The authors should refrain from citing other reviews, which propagates these mistakes (see main comments).

We apologize for the mistake. We read carefully the proposed article that demonstrates how PPARα deletion inhibits ANGPTL4 expression. However, the same article states also PPARγ was investigated. Authors found a reduction in ANGPTL4 expression in mice PPARγ +/- suggesting that different PPARs regulate ANGPTL4 expression. Another paper published in 2020 (PMID: 31163275; ref.#94) found a consistent reduction in secreted ANGPTL4 in liver-specific KO mouse for PPARα. At page 6, from line 269 we stated: “During fasting, ANGPTL4 expression is upregulated in both hepatocyte and adipocytes through activation of PPARs transcription factor (PPARα and PPARγ in particular) in response to glucagon and glucocorticoids [92]–[94].”.

12) The statement that ANGPTL4 is not secreted from hepatocytes is incorrect. Again, the authors refer to a review. It has been shown that human ANGPTL4 can be detected in plasma of hepatocyte humanized mice. Also, fibrates raise plasma levels of ANGPTL4 in humans, supporting the idea that hepatocytes secrete ANGPTL4, which ends up in the circulations. Finally, liver-ANGPTL4 overexpression inhibits The prevailing notion is that liver-derived ANGPTL4 has an endocrine function, in contrast to adipose tissue-derived ANGPTL4.

We carefully checked the literature following the reviewer’s suggestions and we recognize that our statement was incorrect. However, as our focus was to revise the evidences linking ANGPTLs to adipose tissue, we believe not to include many data on the liver secreted ANGPTL4 to avoid confusion. We therefore changed the sentences at page 6, from line 271 as follows: “In the liver, ANGPTL4 is proved to be secreted in fasting and during exercise, probably absolving to endocrine functions [93]. Increased ANGPTL4 expression is associated with increased gluconeogenesis and lipogenesis, which lead to VLDL secretion and fatty-liver in mouse models [88]. In adipose tissue, ANGPTL4 is secreted as a paracrine factor, since it remains bound to ECM surrounding adipocytes [73], [95]. Therefore, PCSKs secretion in response to fasting might be crucial in liberating the N-terminus domain of ANGPTL4, thus enforcing adipose tissue LPL inhibition and directing TGs flux to BAT and muscular tissue for catabolism [88], [96]. (Figure 1)”.

13) The evidence supporting the notion that ANGPTL4 promotes lipogenesis is very weak, certainly not enough to justify figure 1B.

Figure 1 describes a possible functioning ANGPTLs system. Liver steatosis related to ANGPTL4 expression was only observed by Chen et al. in mouse models (Ref. 78). Although weak as evidence, we believe it to be interesting in the panorama of liver – adipose tissue crosstalk. We would suggest maintaining the figure, but highlighting the weakness of this information through a dotted line and stating in the figure legend as follows:” Dotted lines report data with a low level of evidence”. 

14) Figure 1D. Please remove part on ceramide and TNFa. This finding is still contentious and has no direct relevance for the role of ANGPTL4 in WAT during fasting.

The interaction between ANGPTL4 and ceramides is reported in ref. #88 and the one between ceramides and TNFα is in ref. #100. In our opinion, WAT inflammation has a role in determining adipose tissue dysfunction. We suggest to eliminate the interaction in figure 1D since we agree that in physiologic conditions ceramides and TNFα has a little relevance in association with ANGPTL4, whereas we would like maintaining figure 2B as it is, since the interaction may be relevant in conditions of adipose tissue dysfunction. The weakness of this data through a hatched arrow and stated in figure legend:” Dotted lines report data with a low level of evidence”. 

15) Figure 1E and F. The role of ANGPTL4 in BAT should be shown in the context of cold exposure, not fasting.

Thank you for pointing out this issue. We checked literature and found that BAT is indeed active in fasting (PMID: 30262832; ref. 86), however no references were found stating ANGPTLs expression in BAT, associated with feeding and fasting. Therefore, we decided to eliminate panels 1E, 1F and 2C.

Legends were adjusted accordingly.

16) Figure 1 implies that ANGPTL4 protein is produced in the nucleus. Please adjust.

ANGPTL4 was found to be expressed in nucleoplasm by the human protein atlas project. (https://www.proteinatlas.org/ENSG00000167772-ANGPTL4/cell#human;and doi:10.1126/science.aal3321). These findings have been reported at Page 5, from line 223: “Proteomic studies found ANGPTL4 to be also expressed in the nucleoplasm in different cell lines [82], [83]. However, its role in the nucleoplasm is still unknown. “ 

17) Figure 2: please remove link between HIF1a and ANGPTL4. There is no evidence that human WAT is hypoxic during obesity and that hypoxia may influence ANGPTL4 expression during adipose tissue dysfunction in humans.

We checked again the literature and found that HIF may induce expression of ANGPTL4 (PMID:30262832, 21709421; Refs 80-81). We would suggest maintaining the figure. By citing refs 80-81 we have included the evidence in the revised version of the manuscript and therefore, we changed the sentence at Page 8, starting from line 312 as follows: “Permanent block in lipolysis leads to adipocyte mass expansion over oxygen diffusion capacity, this creates adipocyte stress and hypoxia-inducible factor 1-alpha (HIF1α) induction, which in turn induces ANGPTL4 expression [80], [81], together with enhanced circulating glucocorticoids, typical of overnutrition. ANGPTL4 partly blocks WAT LPL, partly induces lipolysis and ceramide production [88].”. We highlighted evidence weakness through a dotted line and stated in figure legend:” Dotted lines report data with a low level of evidence”. 

18) Section 5 needs to be rewritten taking into account that nearly all studies that report on ANGPTL8 in human plasma are wrong due to the use of the non-functional ELISA. These data simply cannot be trusted. Data on plasma ANGPTL4 and ANGPTL3 that used the R&D ELISA can be trusted.

Section 5 was rewritten accordingly. We eliminated ref. 91 which we considered weak. We kept ex ref. 90 (now Ref 100) since the authors did not use a commercial Elisa kit to measure ANGPTL8, but self-developed one. We think this reference might be kept.

We also outlined that serum ANGPTLs level might not be reproducible.

  • In lines 348-350 we stated: ”However, ANGPTLs serum quantification has no international validated standard, and high-grade variability is reported among different publications. Further investigations are needed to assess reproducibility od these data.”.

  • In lines 351-354 sentences have been changed as follows: Levels of circulating ANGPTL8 are apparently crucial actors in impairing glucose metabolism but results are conflicting [71], [85], [105], [106]. In most human studies, serum ANGPTL8 is measured through commercial ELISA kits that showed high inter-, and intra-assay variability. Reported evidence might not be reliable”.

Reviewer 2 Report

This is a nice and concise review about the interplay between adipose tissue, metabolic diseases and Angiopoietins-like proteins, specifically Angptl3, 4 and 8. I only have a few minor concerns:

  1. The authors spent a lot of effort introducing various adipokines secreted by different adipose tissue. However, do these adipokines interplay with Angptls?
  2. The author only mentioned Angptl3/8 DKO mice have intensive beiging and cold exposure may stimulate certain Angptl expression. Please give more information about the interplay between beige adipose tissue and Angptls? Other than fasting and refeeding, how cold regulates Angptls expression.
  3. The review needs careful grammar and spelling correction including figure legends.
  4. Below I just list a few:

Page 3 line 94: β-stimulation should be “β-AR stimulation”. Please revise this sentence as UCP1 mediated uncoupling is uncoupled from ATP formation.

Table 1, please correct spelling “Lipolisis”.

Page 5 line 218: “hampers” should be “hamper”

Author Response

POINT BY POINT REPLY

We thank the reviewer for the careful evaluation of the manuscript, thoughtful considerations and suggestions. Our reply to the points they have raised is reported below. We thank the reviewer for the careful evaluation of the manuscript, thoughtful considerations and suggestions. Our reply to the points they have raised is reported below. (reference to lines are applicable to clear form of revisioned manuscript)

Reviewer 2

This is a nice and concise review about the interplay between adipose tissue, metabolic diseases and Angiopoietins-like proteins, specifically Angptl3, 4 and 8.

We thank the reviewer for having appreciated our manuscript.

I only have a few minor concerns:

1 - The authors spent a lot of effort introducing various adipokines secreted by different adipose tissue. However, do these adipokines interplay with Angptls?

Whole introductory section (1 - 3) was rewritten and reduced. We believe that introducing the role of adipokines and batokines in lipid and glucose metabolism is crucial in order to explain adipose tissue dysfunction.

2-The author only mentioned Angptl3/8 DKO mice have intensive beiging and cold exposure may stimulate certain Angptl expression. Please give more information about the interplay between beige adipose tissue and Angptls? Other than fasting and refeeding, how cold regulates Angptls expression.

Little is known on the role of cold in inducing ANGPTLs. The idea beneath the manuscript is that DKO of ANGPTL3 and ANGPTL8 determines excessive chylomicrons clearance and metabolic shift towards lipids catabolism, hence temperature rise. However, there is no strong evidence linking cold exposure and specific ANGPTLs expression in WAT or BAT. One paper evaluated changes in plasma ANGPTLs in lean and obese human, however ELISA techniques are not yet sufficiently reliable to draw specific conclusions. Published matter from hypermetabolic DKO mouse is stated at page 5, starting from line 235: “Banfi et al. [85] found that knockout mouse models for both ANGPTL3 and ANGPTL8 had intense adipose tissue beiging in comparison with their wild-type littermates. In particular, WAT of double KO mice expressed higher levels of UCP1, a typical BAT marker [85]. KO mice were also hypermetabolic, raising their basal temperature of 1°C during feeding [85]. A similar effect could be evoked during fasting by the administration of a β3-agonist, or suppressed with a β3-antagonist during feeding [85]. It is still unknown how ANGPTL3 and ANGPTL8 may regulate the expression of UCP1. However, modification in levels of circulating lipids is possibly involved in enhancing β3-adrenergic receptor functioning, which is the main player in steering WAT beiging [30], [31], [85], [86].”

3-The review needs careful grammar and spelling correction including figure legends.

We thank the reviewer. English grammar has been revised, including figure legends.

Below I just list a few:

Page 3 line 94: β-stimulation should be “β-AR stimulation”. Please revise this sentence as UCP1 mediated uncoupling is uncoupled from ATP formation.

Thank you, sentence has been revised.

Table 1, please correct spelling “Lipolisis”.

In order to reduce sections 1-3, and to focus attention in ANGPTLs functions, Table 1 was eliminated.

Page 5 line 218: “hampers” should be “hamper”

The correction has been done.

Round 2

Reviewer 1 Report

The authors have made appropriate changes to the manuscript. Despite that, the manuscript still requires some English language editing. Below, a number of corrections and comments are specified.

Line 43: “Adipose tissue encompasses for 3 tissues that differ for development, anatomical location, and metabolism: the white adipose tissue (WAT), the brown adipose tissue (BAT), and the beige adipose tissue.”. Change to “Adipose tissue encompasses 3 tissues that differ in the development, anatomical location, and metabolism: the white adipose tissue (WAT), the brown adipose tissue (BAT), and the beige adipose tissue.”.

Line 48: “Despite WAT has been considered ...”. Change to “Despite the fact that WAT has been considered...”.

Line 89: ”... as well as cytokines such IL-6,...”. Change to “... as well as cytokines such as IL-6,...”

Line 91: “In mouse models, BAT transplant determined increased...”. Change to “In mouse models, BAT transplant led to increased...”.

Line 123: “...it must be recognized that little expressed proteins might not...”. Change to “...it must be recognized that weakly expressed proteins might not...”.

If the author are aiming to write a balanced review (which I hope), they are strongly encouraged to specifically mention the conflicting evidence on hypoxia in obese human adipose tissue (see PMID: 21670228, 31077538).

Line 173. It is perhaps a bit exaggerated to refer to ANGPTL3, ANGPTL4, and ANGPTL8 as the main regulators of triglyceride and energy metabolism.

Authors are strongly encouraged to be very critical of the results of the protein Atlas. The antibodies used have not necessarily been properly validated and the results should be taken with a lot of caution. Accordingly, the authors should remove pancreas as a site of expression of ANGPTL4.

Line 201. Change inductor into inducer.

Line 212-218. There is also evidence that ANGPTL4 promotes the cleavage of LPL. This would be relevant to include.

Line 271. Absolve. Poor word choice.

Author Response

POINT BY POINT RESPONSE – SECOND ROUND REVISION

The authors have made appropriate changes to the manuscript. Despite that, the manuscript still requires some English language editing. Below, a number of corrections and comments are specified.

 We thank the reviewer for appreciating the last version of our manuscript. The text was revisioned again for grammar and punctuation, with the help of  an English editing software.

Line 43: “Adipose tissue encompasses for 3 tissues that differ for development, anatomical location, and metabolism: the white adipose tissue (WAT), the brown adipose tissue (BAT), and the beige adipose tissue.”. Change to “Adipose tissue encompasses 3 tissues that differ in the development, anatomical location, and metabolism: the white adipose tissue (WAT), the brown adipose tissue (BAT), and the beige adipose tissue.”.

Thanks for highlighting the grammar error. The sentence was changed accordingly.

Line 48: “Despite WAT has been considered ...”. Change to “Despite the fact that WAT has been considered...”.

Thanks for highlighting the grammar error. The sentence was changed accordingly.

Line 89: ”... as well as cytokines such IL-6,...”. Change to “... as well as cytokines such as IL-6,...”

Thanks for highlighting the grammar error. The sentence was changed accordingly.

Line 91: “In mouse models, BAT transplant determined increased...”. Change to “In mouse models, BAT transplant led to increased...”.

Thanks for highlighting the grammar error. The sentence was changed accordingly.

Line 123: “...it must be recognized that little expressed proteins might not...”. Change to “...it must be recognized that weakly expressed proteins might not...”.

Thanks for highlighting the grammar error. The sentence was changed accordingly.

If the author are aiming to write a balanced review (which I hope), they are strongly encouraged to specifically mention the conflicting evidence on hypoxia in obese human adipose tissue (see PMID: 21670228, 31077538).

We agree that our review should be balanced. We added suggested literature as ref. #45 and #46. Sentence from Line 155 was changed accordingly: “Conversely, in obese people, adipose depots expand beyond the tissue’s capacity of adequate angiogenesis, possibly resulting in persistent hypoxia, fibrosis, and adipocyte death [1], [43], [44]. However, studies on human adipose tissue oxygenation in dysfunctional conditions are still controversial: while some authors found SAT to be hypoxic in obese patients [43], [44], others found it to be hyperoxic [45], [46].”.

Line 173. It is perhaps a bit exaggerated to refer to ANGPTL3, ANGPTL4, and ANGPTL8 as the main regulators of triglyceride and energy metabolism.

We agree it might be exaggerated. The sentence in line 179 was changed as follows:” ANGPTL3, ANGPTL4, and ANGPTL8 are essential regulators of triglyceride and energy metabolism. In this paragraph, we will focus on their regulatory function on adipose tissue metabolism and dysfunction [10]–[12].”.

Authors are strongly encouraged to be very critical of the results of the protein Atlas. The antibodies used have not necessarily been properly validated and the results should be taken with a lot of caution. Accordingly, the authors should remove pancreas as a site of expression of ANGPTL4.

We agree protein atlas is not the best of evidence, we followed your advice and removed pancreas as a site of expression of ANGPTL4. Moreover, we added in line 234: “However, its possible role in the nucleoplasm is still unknown, and its cellular localization should be further confirmed.”

Line 201. Change inductor into inducer.

Thanks for highlighting the grammar error. The sentence was changed accordingly.

Line 212-218. There is also evidence that ANGPTL4 promotes the cleavage of LPL. This would be relevant to include.

We Agree that adding ANGPTL4 as promoter of LPL cleavage may be relevant. We added ref. #80 and changed text from line 228:” ANGPTL4 was also found to promote intracellular cleavage of LPL mediated by PCSKs in adipose tissue of murine models [80]. ANGPTL3 and ANGPTL8 might have similar inhibition patterns.”.

Line 271. Absolve. Poor word choice.

We agree that absolve might not have been the best word choice, sentence in line 290 was changed accordingly as follows: “In the liver, ANGPTL4 is proved to be secreted in fasting and during exercise, probably carrying out endocrine functions [96].”